# Sarilumab Administration in COVID-19 Patients: Literature Review and Considerations

Andrea Marino [1,2], Antonio Munafò [3,*], Egle Augello [3], Carlo Maria Bellanca [3], Carmelo Bonomo [1], Manuela Ceccarelli [2], Nicolò Musso [1], Giuseppina Cantarella [3], Bruno Cacopardo [2] and Renato Bernardini [3,4]

1   Department of Biomedical and Biotechnological Science (BIOMETEC), University of Catania, 95123 Catania, Italy; andreamarino9103@gmail.com (A.M.); carmelo.bonomo@phd.unict.it (C.B.); nmusso@unict.it (N.M.)
2   Unit of Infectious Diseases, Department of Clinical and Experimental Medicine, ARNAS Garibaldi Hospital, University of Catania, 95123 Catania, Italy; manuela.ceccarelli86@gmail.com (M.C.); cacopard@unict.it (B.C.)
3   Department of Biomedical and Biotechnological Science, Section of Pharmacology, University of Catania, 95123 Catania, Italy; egle.augello@gmail.com (E.A.); uni318437@studium.unict.it (C.M.B.); gcantare@unict.it (G.C.); bernardi@unict.it (R.B.)
4   Unit of Clinical Toxicology, Policlinico G. Rodolico, School of Medicine, University of Catania, 95123 Catania, Italy
*   Correspondence: uni315775@studium.unict.it

**Abstract:** Two years have passed since WHO declared a pandemic state for SARS-CoV-2 infection. COVID-19 pathogenesis consists of a first viral phase responsible for early symptoms followed by an inflammatory phase, cytokine-mediated, responsible for late-onset manifestations up to ARDS. The dysregulated immune response has an outstanding role in the progression of pulmonary damage in COVID-19. IL-6, through the induction of pro-inflammatory chemokines and cytokines, plays a key role in the development and maintenance of inflammation, acting as a pioneer of the hyper-inflammatory condition and cytokine storm in severe COVID-19. Therefore, drugs targeting both IL-6 and IL-6 receptors have been evaluated in order to blunt the abnormal SARS-CoV-2-induced cytokine release. Sarilumab, a high-affinity anti-IL-6 receptor antibody, may represent a promising weapon to treat the fearsome hyperinflammatory phase by improving the outcome of patients with moderate-to-severe COVID-19 pneumonia. Further prospective and well-designed clinical studies with larger sample sizes and long-term follow-up are needed to assess the efficacy and the safety of this therapeutic approach to achieve improved outcomes in COVID-19.

**Keywords:** COVID-19; cytokines; IL-6; IL-6R; sarilumab

## 1. Introduction

Almost two years have passed since the WHO declared the SARS-CoV-2 outbreak a global pandemic [1]. Up to 1 April 2022, >490 million cases of SARS-CoV-2 infections were reported, and >6 million individuals succumbed to the disease worldwide. To date, a total of 14 million cases have been reported in Italy, 160,000 of which have not survived [2].

It is clear now that uncontrolled systemic inflammation represents a critical element in the progression of COVID-19 to acute respiratory distress syndrome (ARDS) [3].

This non-specific and deleterious inflammatory response seems to lead to alveolar damage as a result of inflammatory cell infiltration, pulmonary edema, and endothelial impairment, along with microvascular thrombosis, playing a key role in the development of severe COVID-19 [4]. As a matter of fact, dexamethasone treatment is correlated with better outcomes in patients with severe COVID-19 [5,6].

Interleukin (IL)-6 cascade has already been proposed as a potential target for immunomodulatory therapy against moderate systemic hyper-inflammation during SARS-CoV-2 infection [7]. Scientific literature and international guidelines suggest the use of tocilizumab, a recombinant monoclonal antibody, in addition to standard of care (SOC),

due to its ability to lower the risk of respiratory deterioration, thus reducing mortality [8]. When tocilizumab is not available, sarilumab, a human monoclonal antibody targeting IL-6 soluble receptors, which is already approved for rheumatoid arthritis (RA) treatment, represents a valid alternative for IL-6 blockade [9].

This review aims to report the most recent information and available data on the role of sarilumab as a potential drug for the management of COVID-19.

## 2. Considerations concerning IL-6, IL-6R, and Sarilumab

IL-6 is a small glycoprotein with well-defined, context-dependent pro- and anti-inflammatory properties that acts as a critical signaling node, transmitting defense signals from pathogen invasion or tissue damage sites [10]. IL-6 systemic physiologic functions include the stimulation of acute-phase protein production, the activation of T-cells, the induction of differentiation, and the proliferation of hematopoietic stem cells and hepatic cells [11]. However, excessive and dysregulated production of IL-6 has a pathological effect on chronic inflammation and autoimmunity [12]. In humans, IL-6 gene functional regulatory elements are represented by the nuclear factor kappa B (NF-kB) binding site, nuclear factor IL6 (NF-IL6), specificity protein 1 (SP1), cyclic AMP response element binding protein (CREB), interferon regulatory factor 1 (IRF-1), and activator protein 1 (AP-1). Among these, NF-kB is one of the main transcriptional factors commonly activated, either by Toll-like receptor (TLR)-mediated signals or by pro-inflammatory cytokines, including tumor necrosis factor $\alpha$ (TNF $\alpha$), IL-1, and IL-17 [13,14]. IL-6 mRNA is regulated at the post-transcriptional level by adenylate-uridylate rich elements (AREs) located in the 3′UTR region of IL-6 mRNA. A nuclease known as regulatory RNase-1 (Regnase-1) degrades transcriptionally active IL-6 mRNA by binding with IL-6 3′UTR in the cytoplasm, endoplasmic reticulum, and ribosomes [15]. Furthermore, inactive IL-6 mRNA is degraded by roquin, another RNA-binding protein, in stress granules and processing bodies. In contrast, an RNA-binding protein, AT-rich interactive domain-containing protein 5a (Arid5a), is expressed in response to the presence of lipopolysaccharides (LPS), IL-1, and IL-6 in macrophages and T helper (Th) 17 cells. Arid5a is imported into the nucleus via the importin-$\alpha$/$\beta$1 pathway and binds to the 3′UTR of IL-6 mRNA. Afterwards, Arid5a exports IL-6 mRNA to the cytoplasm via the chromosomal region maintenance 1 (CRM1) pathway by binding to up-frameshift protein 1 (UPF1). The IL-6 mRNA bound to Arid5a is protected from degradation by Regnase-1 in the cytoplasm [16]. The counteracting roles of Arid5a and Regnase-1 regarding the stability of IL-6 mRNA indicate that their balance regulates IL-6 production [17]. Thus, this process follows a fine and complex regulation at the transcriptional and post-transcriptional levels, a complexity that reflects the homeostatic role of this cytokine.

The pleiotropic function of IL-6 finds its realization through interaction with its unique receptor system. IL-6 effects begin by attaching to the IL-6 receptor (IL-6R), also designated as CD126, which is mainly expressed on the surface of hepatocytes and hematopoietic cells, including T-cells, activated B-cells, monocytes/macrophages, and neutrophils [11]. The attachment of IL-6 to membrane-IL-6-R (mIL-6R) specifically determines the activation of a signaling pathway named cis-signaling or the classis mode that mediates host protection against intracellular pathogens and tissue homeostasis by inducing macrophage M2 polarization (anti-inflammatory) and by eliciting the clearance of remnants through the hepatic release of opsonins [18]. IL-6R also exists in a soluble form (sIL-6R) [19], which is the result of either the limited proteolysis of mIL-6R by a metalloprotease-17 (ADAM-17) or the protein translation from an alternatively spliced mRNA [17]. sIL-6R is expressed in serum and synovial fluid, inducing a variety of cells that are unable in any other way to respond to IL-6 stimulus due to the lack of constitutive receptor expression [20]. The complex IL-6/sIL-6R stimulates cells through a process known as trans-signaling, which drives the pro-inflammatory activation of pneumocytes, adipose tissue-associated macrophages, neutrophils, and endothelial cells [21]. In particular, the pro-inflammatory role of IL-6 in viral infection is exerted by mononuclear recruitment, the inhibition of T-cell apoptosis,

the stimulation of acute-phase protein production, and the inhibition of T regulatory cell differentiation [22]. A third signaling modality is trans-presentation, in which IL-6 first binds to mIL-6R of one cell and then the complex binds to gp130 in T-cells, promoting T-cell differentiation into pathogenic Th 17 cells [19]. In summary, cis-signaling is important for regenerative and protective functions, along with the homeostatic role, whereas trans-signaling and trans-presentation are responsible for the pro-inflammatory activity of this pleiotropic cytokine [11].

The effective binding between the cytokine and its receptor requires the association with signal-transducing β-subunit glycoprotein 130 (gp130), also known as CD130, to activate the signaling cascade [23]. Coupling of IL-6 with IL-6R initiates gp130 homodimerization that results in a conformational change responsible for the activation of gp130-associated Janus family tyrosine kinases (JAKs), which, in turn, phosphorylates 5-tyrosine residues located on the cytoplasmic portion of gp130. These membrane-proximal tyrosines stimulate the mitogen-activated protein kinase (MAPK) pathway through the phosphatase SHP2 and the phosphorylation of other tyrosine residues leading to the recruitment of signal transducers and activators of transcription (STAT) factors [24]. Afterwards, STAT complex–proteins travel into the nucleus, where they stimulate the expression of target genes, including proliferative, anti-apoptotic, and acute-phase protein genes [25]. The interaction between IL-6 and its soluble receptor leads to a complex that allows the activation of cells that express gp130, resulting in the cis-signaling transduction pathway. Because each type of cell expresses gp130, the complex IL-6/sIL-6R could potentially prompt significant inflammatory dysregulation. To avoid this eventuality, a soluble and non-functional form of gp130 (sgp130) is expressed as a buffer to neutralize IL-6 activity, effectively competing with membrane-bound gp130 [26].

The knowledge about the molecular mechanisms underlying the interaction between IL-6 and its receptor has allowed the development of monoclonal antibodies either directed against the IL-6 receptor (tocilizumab and sarilumab) or IL-6 cytokine (sirukumab, olokizumab, and clazakizumab).

Among these, we focused our attention on sarilumab, a fully human IgG1 monoclonal antibody (mAb) produced in Chinese Hamster Ovary cells by recombinant DNA technology. This mAb binds both soluble and membrane-bound IL-6R with high affinity, interfering with cis- and trans-IL-6-mediated inflammatory pathways. Inhibition of IL-6 signaling interrupts the cytokine-mediated cascade, with no evidence of complement- or antibody-dependent cell-mediated cytotoxicity [27].

Sarilumab prevents IL-6-induced effects in cells expressing mIL-6R and gp130. Therefore, the mechanism of action is identical to tocilizumab, but sarilumab presents a lesser dissociation constant and a higher capacity and affinity for IL-6R [9]. Moreover, it stops the trans-signaling pathway mediated by the IL-6/sIL-6R complex in cells expressing only gp130, and it does not show activity if IL-6 is absent [28] (Figure 1).

Sarilumab received FDA approval on 22 May 2017 for the treatment of adult patients with moderate-to-severe RA who show inadequate response to other disease-modifying anti-rheumatic drugs (DMARDs). Additionally, it is currently under investigation for the treatment of other hyperinflammatory conditions [9]. Sarilumab can be used either in monotherapy (in case of intolerance to methotrexate or when treatment with methotrexate is inappropriate) or in combination with conventional DMARDs [29].

After SARS-CoV-2 infection, immune cells release cytokines, including IL-6. IL-6 can either attach to its respective membrane-bound receptor (mIL-6R) or soluble receptor (sIL-6R), which activates JAK/STAT pathways. The different signaling modalities, including cis-signaling, trans-signaling, and trans-presentation are reported. Abbreviations: gp130, glycoprotein 130; IL-6, interleukin-6; JAK, Janus family tyrosine kinase; mIL-6R, membrane-bound interleukin-6 receptor; sIL-6R, soluble interleukin-6 receptor; STAT, signal transducer and activator of transcription; P, phosphoryl group. Illustrations use elements from Servier Medical Art (https://smart.servier.com/, accessed on 10 April 2022).

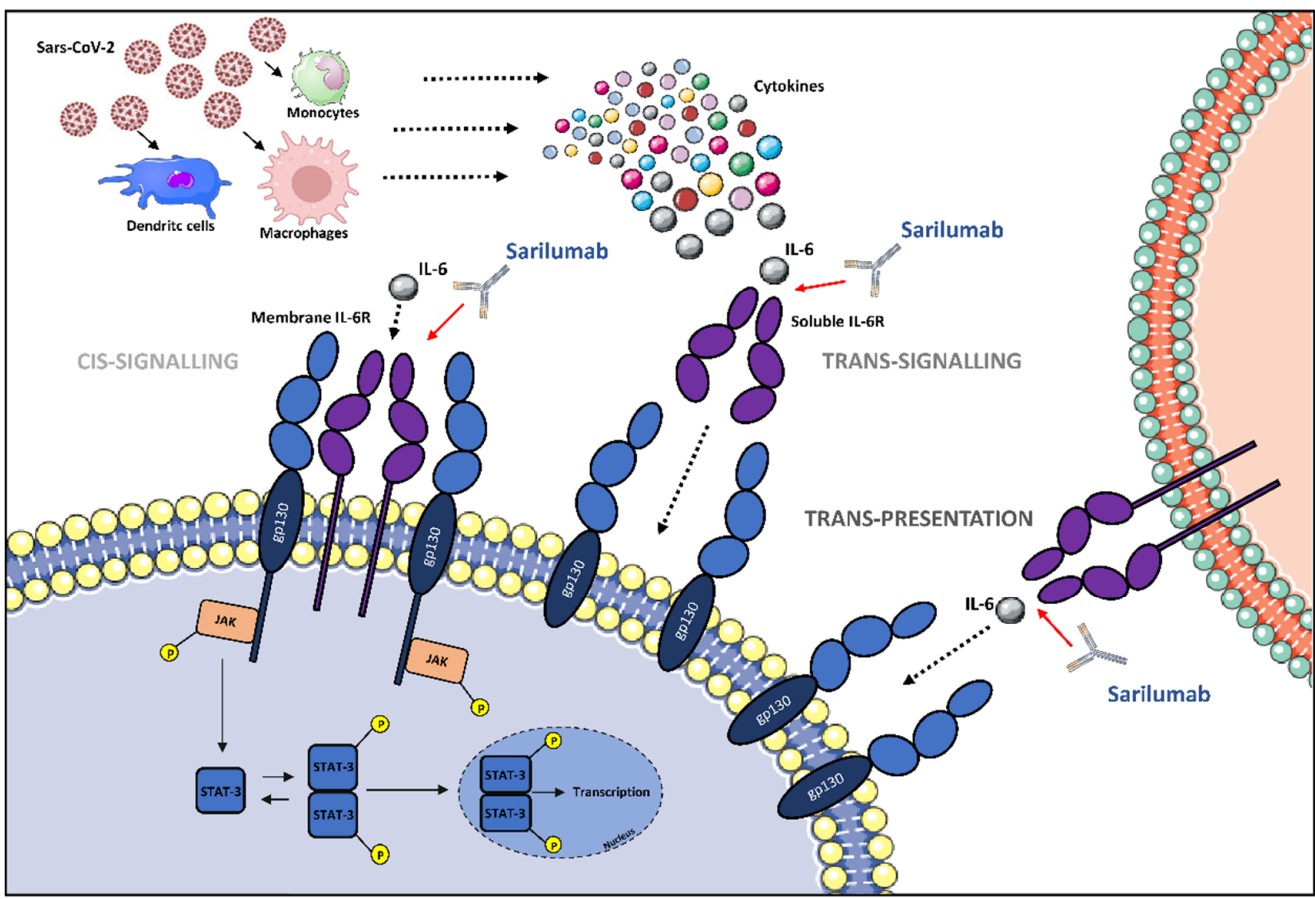

**Figure 1.** The potential role of IL-6 receptor antagonization by sarilumab in SARS-CoV-2-induced cytokine release syndrome (CRS)-like syndrome.

### 3. Cytokine Storm and Rationale for IL-6 Inhibitor Administration in COVID-19 Patients

Critical and severe COVID-19 forms, constituting 2–5% and 15% of cases, respectively, are assumed to occur from SARS-CoV-2-induced autoinflammatory syndrome, both at systemic and pulmonary levels, in which a dysregulated immune response, due to overflowing cytokine production and release, leads to widespread tissue and vascular damages [30]. Viral infections and autoimmune pathologies are known to be correlated to an abnormal immune response known as a "cytokine storm", characterized by an excessive release of pro-inflammatory cytokines [31,32]. Host tissue toxicity, multiple-organ failure, and high fever are the most common effects of this syndrome, which could result in a fatal outcome. However, dysregulated inflammatory response may involve every organ, including skin, which is considered by some authors as a "sentinel" of early COVID-19 manifestations [33–35].

As well as SARS-CoV and MERS-CoV, SARS-CoV-2 could be responsible for the development of a hyperinflammatory state, worsening patients' clinical conditions and outcomes. Cytokines play a central role in the pathogenesis of COVID-19, emerging both as useful biomarkers in predicting disease severity and as strategic targets for therapy [36].

Different clusters of cytokines are selectively expressed according to disease stages [36]. Namely, evidence has revealed that severe COVID-19 patients admitted to intensive care units (ICUs) show elevated levels of pro-inflammatory mediators such as IL-1, IL-2, IL-6, IL-7 IFN-$\gamma$, and TNF-$\alpha$ and reduced lymphocyte counts when compared to non-ICU patients [37,38]. Among these, IL-6 was suggested as the key player in the cytokine storm observed during COVID-19 pathogenesis [39–41]. Systemic and lung IL-6 levels progressively increase in COVID-19 patients with disease severity, reaching the peak

in the critical ones, the results being typically associated with lymphopenia, systemic inflammation, hypoxemia, and unfavorable prognosis [42]. Considering the factors that most affect the production of IL-6 in COVID-19 disease, an important role is played by the downregulation of the SOC3 pathway, which has negative feedback on IL-6 production [43]. Alternative pathways proposed for the increased production of IL-6 include the inactivation of Regnase-1, a known IL-6 inhibitor, and angiotensin-2-increased NFKB gene expression by TNF-$\alpha$ [44].

IL-6 significantly contributes to immune dysregulation in COVID-19 by acting in two main pathways: on the one hand, this cytokine may lead to a dysfunction of natural killers (NK) cells and cytotoxic CD8$^+$ T-cells, suppressing antiviral defenses [45]; on the other hand, it may stop the differentiation of T reg cells, raising Th 17 polarization of CD4$^+$ T-cells, thus leading to unrestrained hyperinflammation [46]. In the final stages, these mechanisms could result in a macrophage activation syndrome (MAS)-like syndrome, characterized by lymphocyte exhaustion along with aberrant immune response, vascular leakage, coagulopathy, and ARDS, up to multi-organ failure [47].

The IL-6 central role in the pathogenesis of SARS-CoV-2 infection is supported by the evidence that elevated IL-6 levels correlate with lymphopenia, higher viral load, hypoxemia, systemic inflammation, and poor outcomes [48], and it is confirmed by the fact that polymorphisms inactivating the IL-6 receptor gene have been revealed to be protective against COVID-19 progression, reducing the risk of hospitalization in these patients [49]. Based upon this evidence, therapeutic blockade of IL-6 signaling may constitute an efficient strategy to prevent respiratory function deterioration during COVID-19, reducing overall mortality in these subjects.

As regards signaling modalities, trans-signaling and trans-presentation are suggested to be the main pathogenic pathways in severe progressive COVID-19, accounting for disseminated inflammation up to shock, secondary to cytokine-mediated dysfunction [50], whereas IL-6 cis-signaling, exerting negative feedback mechanisms on pro-inflammatory cytokines, appears as predominant in the advanced stages of disease in which the IL-6 peak is accompanied by elevated IL-10 concentrations, endeavoring to restore homeostatic conditions [51].

Therefore, a therapeutic window designed to block IL-6 signaling is targeted to trans-signaling and trans-presentation pathways. Literature evidence shows that interfering with the IL-6 cascade, both in the early stage of COVID-19, in which the cytokine could have protective functions, and during advanced stages, when IL-6 cis-signaling regulates and promotes cells growth and survival, may not be useful or may even worsen the disease's clinical evolution [52].

In conclusion, therapeutic blockade of IL-6 pathways should not be performed in critically ill patients, in whom IL-6 cis-signaling is predominant with favorable homeostatic roles, nor in the early (viral) phase of SARS-CoV-2 infection, in which IL-6 signaling may be useful to block viral replication and dissemination [30]. The proven benefits of anti-IL-6 treatment in COVID-19 patients could be achieved by disrupting only pro-inflammatory IL-6 trans-presentation and trans-signaling, prevalent and pathogenic in severe disease, by acting no later than the second week of symptom onset (or within seven days of hospitalization) [30].

Although it is often cumbersome to stratify and recognize the right patients to treat with anti-IL-6 agents within the right time of COVID-19 pathophysiology, some important inflammatory and respiratory parameters have been suggested to help clinicians manage the decision. Along these lines, Zizzo et al. proposed that IL-6 levels between 35 and 90 pg/mL, as well as C-reactive protein (CRP) levels between 120 and 160 mg/L, may help to identify those severe patients who would benefit from the cytokine cascade interruption within 6 days of hospitalization [30]. In the same way, COVID-19 patients with serum LDH levels > 550 U/L, ferritin levels > 1600–2000 ng/mL, and D-dimer levels > 3000–5000 ng/mL represent critically ill patients who would not respond to anti-IL-6 treatments [30].

Considering respiratory parameters, a $PaO_2/FiO_2$ ratio between 200 and 100 mmHg could stratify the right cohort of patients to treat with IL-6 inhibitors, whereas patients with values < 100, recognized as critically ill, and those with values > 200, stratified as mild–moderate, would no achieve benefits from anti-IL-6 therapies [30,53].

## 4. Literature Evidence

Up to now, SARS-CoV-2 infection pathophysiology remains mostly unclear, arguably as it is not attributable only to the virus; in fact, both immune and inflammatory responses seem to play a key role in disease development and duration, especially as regards its severe form.

In order to avoid disease progression, we could intervene during the first phase of infection (viral phase) with some antiviral drugs or monoclonal antibodies, administering them within the correct time lapse. However, it is the second phase (inflammatory phase) that challenges clinicians because of its severity, complexity, and lack of standard treatments. Several studies have reported that COVID-19 causes the so-called Cytokine Release Syndrome (CRS), leading to a dysregulated immune response up to ARDS. CRS is considered as an immune system overreacted reaction developing in an unregulated manner, the same pathogenetic mechanism which underlines autoimmune and hematologic diseases [54,55]. As confirmation, drugs to treat this phase are also administered against autoimmune disorders [54].

Various pro-inflammatory cytokines have been investigated as the cause of CRS; among them, IL-6 is one of the most studied due to its importance in inflammatory pathways.

IL-6 has been assessed as both an inflammatory/prognostic marker (since its levels correlate with the inflammation state) and as a therapeutic target. Monoclonal antibodies targeting IL-6 and IL-6 receptors are recommended by Italian and American guidelines to treat patients with severe and critical COVID-19 [56,57].

Sarilumab, a humanized mAb (IgG1 subtype), specifically binding both mIL-6R and soluble sIL-6R, inhibits IL6-mediated pathways involving glycoprotein 130 (gp130) along with STAT-3, a signaling transducer and transcription activator. Sarilumab has been investigated in a small number of studies, the results of which were not conclusive. Della-Torre et al. [58] reported data on 28 COVID-19 patients treated with a single dose of sarilumab iv that showed a decrease in recovery time but without statistically significant differences in terms of mortality and overall improvement between patients treated with standard SOC. Gremese et al. [59] studied 53 patients treated with sarilumab, almost all of whom received a second infusion; 14 of them were from ICUs and showed an improvement in clinical conditions along with a reduction in oxygen supplementation therapy; in addition, more than half of the ICU patients showed clinical amelioration after sarilumab administration. Although on a smaller sample size, the same results were shown by Benucci et al. [60] in 8 patients treated with sarilumab.

To date, only a few randomized controlled trials have been published about sarilumab administration in COVID-19 patients, and no specific meta-analyses have yet been performed. The WHO Rapid Evidence Appraisal for COVID-19 Therapies (REACT) working group performed a prospective meta-analysis of 10,930 patients participating in 27 clinical trials, identifying a lower 28-day, all-cause mortality of 22% for patients treated with IL-6 antagonists compared with 25% in a placebo group [61]. In the study performed by the REMAP-CAP collaborative group [62], 48 patients were assigned to one dose of 400 mg sarilumab iv administration; the results showed that sarilumab improved in-hospital survival compared with usual care. A larger study, performed by Lescure et al. [63] on 420 subjects, did not demonstrate the efficacy of sarilumab as regards outcome and survival rates in patients hospitalized with severe COVID-19 and receiving supplemental oxygen despite improved recovery time.

The CORIMUNO19 group performed an open-label, randomized, controlled trial with 148 patients randomly assigned to sarilumab or SOC, with half of the patients in the sarilumab group treated with a second dose [64]. This trial did not highlight any effect of

sarilumab in patients with moderate to severe COVID-19 in terms of mortality rate nor the decreasing proportion of patients needing non-invasive ventilation.

Dexamethasone administration was highly variable across studies on sarilumab therapy in contrast to trials about tocilizumab administration, in which the anti-IL-6 drug was almost always administered together with corticosteroids. This difference might explain the diverse results between tocilizumab and sarilumab in favor of the former drug in previous studies; however, these data should be clarified in larger trials.

As regards the scientific literature on sarilumab adverse drug reactions, although with some limitations (absence of a control group, single-center setting, concomitant treatments), Gremese et al. [59] did not register any serious adverse events or secondary infections related to the treatment with sarilumab. In the study carried out by Lescure et al. [65], the occurrence of adverse events of different severity was similar between both the treatment group and the placebo group. No serious adverse events were reported in the REMAP-CAP study [62], and the CORIMUNO19 group [64] reported a few cases of temporary neutropenia, which is a common side effect of all IL-6 blockers. In the same study, a non-statistically significant increased number of bacterial infections was reported in the sarilumab group (12 patients) compared to the control group (7 patients). Although transient neutropenia has been observed in several studies involving patients affected by autoimmune diseases, such as RA treated with drugs targeting the IL-6 cascade, the serious infection rate in these patients did not appear to be increased, suggesting that blocking IL-6 pathways may influence the neutrophils' number without compromising their function [66].

Wright et al., observing in vitro the effect of anti-IL-6 drugs on the neutrophil population, showed that anti-IL-6-induced-neutropenia is not directly determined by increased neutrophil apoptosis.

There are no definitive data about neutrophil count reduction after IL-6 blockade, only several hypotheses supported by scientific literature [67]. Decreased neutrophil counts may be the result of different cell distributions between circulating and marginating pools due to IL-6-blocking drug effects on L-selectin and P- selectin ligand expression on the neutrophils' surface. Because of the role of IL-6 in accelerating neutrophils' release from the bone marrow into circulation, anti-IL-6 drugs may provoke an increase in transit time, causing transient neutropenia [67,68]. Significantly, literature data revealed that while total neutrophil count may decline after anti-IL-6 drugs, the remaining neutrophils are totally functional without impairment in their ability to mount a respiratory burst or phagocytose bacteria. Moreover, the transient nature of neutropenia suggests that neutrophil counts begin to resolve within days, minimizing the risk of serious infection [68].

WHO-conducted metanalysis [61] showed an increased risk of infection for patients treated with IL-6 receptor antagonists compared with those treated with SOC or placebo, and consistent results were disclosed by Han et al. [69] in a meta-analysis investigating infection risk during the use of IL-6 drugs. However, considering patients with severe and critical COVID-19, no statistical significance was highlighted for the overall risk of secondary infections due to IL-6 antagonists' treatments [70,71]. Clearly, before IL-6 antagonist administration, viral and bacterial infections, especially latent or immunosuppressing infections, should be ruled out [72–74].

Besides neutropenia, along with the relative risk of bacterial infections, other possible sarilumab adverse reactions may be represented by increased transaminase levels without clinically evident hepatic injury, increased total cholesterol levels, especially low-density lipoprotein levels, and injection site reactions (erythema or pruritus) [75].

Moreover, sarilumab could enhance the activity of cytochrome CYP3A4, which is inhibited by the IL-6 cascade, resulting in decreased serum concentrations of other drugs [29]. Due to that, prior to sarilumab administration, drug–drug interaction should be carefully evaluated [76].

Furthermore, although results from several retrospective and prospective studies did not show an increased risk of infection after IL-6 antagonist treatment in COVID-19 patients,

the most common infections reported are represented by Gram-positive bacteremia [77,78], which occurred more often than Gram-negative bacteremia, fungemia, or viremia, [79,80]; *Staphylococcus aureus* was the more frequent causative organism [81].

## 5. Conclusions

The beneficial effects of IL-6 inhibitors, particularly anti-IL-6 mAb, in the management of COVID-19 have long been debated owing to discrepancies in study results due to heterogeneity in sample size, patient series composition, treatment protocols, concomitant therapies, and disease severity. As a matter of fact, prior studies with negative results recruited fewer than 100 patients in the treatment arm, while the best outcomes were achieved in the largest and most recent trials, suggesting that the small size and composite primary endpoints of early trials were probably underpowered to detect conclusive results. Based on available data, patients with an early hyperinflammatory phenotype and minimal evidence of organ damage within the second week of symptom onset may mostly benefit from IL-6 pathway blockade. By contrast, IL-6 inhibition no longer appears to be useful in critically ill patients since organ damage has already occurred and, consequently, clinical benefits are blunted. The distinct roles of IL-6 in relation to disease stages and the biological significance of clinical patterns and serological markers become essential in order to identify and summarize the baseline characteristics of patients who should best respond to these therapies. Relatively simple items, including $PaO_2$, CRP, ferritin, and D-dimer, could actually reflect the complexity and heterogeneity of disease biology and might also be helpful in identifying patients progressing toward severe to critical stages. A deeper understanding of the quantitative and temporal variations in cytokine pathways is, therefore, crucial to choosing the proper therapeutic window. In summary, sarilumab is a safe drug with good clinical outcomes in patients with COVID-19 and, hence, could be an alternative regimen for the treatment. Further prospective and well-designed clinical studies with larger sample sizes and long-term follow-up are needed to assess the efficacy and safety of this therapeutic approach to achieve improved outcomes in COVID-19.

**Author Contributions:** Conceptualization, R.B., G.C. and B.C.; writing—original draft preparation, A.M. (Andrea Marino) and A.M. (Antonio Munafò); writing—review and editing, C.M.B., E.A. and C.B.; M.C. gave clinical assistance; image preparation, N.M. All authors have read and agreed to the published version of the manuscript.

**Funding:** This research received no external funding.

**Institutional Review Board Statement:** Not applicable.

**Informed Consent Statement:** Not applicable.

**Data Availability Statement:** No new data were created or analyzed in this study. Data sharing is not applicable to this article.

**Conflicts of Interest:** The authors declare no conflict of interest.

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
