# Peer review of "Sarilumab Administration in COVID-19 Patients: Literature Review and Considerations"

_2036-7449, doi:10.3390/idr14030040_

Round 1

Reviewer 1 Report

In the manuscript "Sarilumab administration in COVID-19 patients: literature review and considerations", the authors evaluated the efficacy of Sarilumab in patients with SARS-CoV-2 infection. They analyzed data reported in the literature, confirming that Sarilumab could be an alternative regimen for the treatment of COVID-19. The work is quite interesting but I want to suggest reviewing the same citations reported in the bibliography.

Author Response

Dear Editor,

We appreciate you and the reviewers for giving us the opportunity to submit a revised draft of the manuscript and for your precious time in reviewing our paper.

We have carefully considered the comments and tried our best to address every one of them. We hope the manuscript after meticulous revisions meet your high standards. Additional constructive comments, if any, are welcomed. Below we provide the point-by-point responses.

All modifications in the manuscript have been highlighted with “Track changes” option on MS Word.

Reviewer 1

In the manuscript "Sarilumab administration in COVID-19 patients: literature review and considerations", the authors evaluated the efficacy of Sarilumab in patients with SARS-CoV-2 infection. They analyzed data reported in the literature, confirming that Sarilumab could be an alternative regimen for the treatment of COVID-19. The work is quite interesting but I want to suggest reviewing the same citations reported in the bibliography.

Reply: Thank you for your valuable opinion and consideration. We tried to summarize the main evidence about the subject reviewing the last scientific papers. We revised the reference section according to both your and other reviewers’ suggestions.

Reviewer 2 Report

Dear Publisher and Authors,
I spent a lot of time reviewing this manuscript as, before proceeding with the check, I wanted to study and compare the most up-to-date treatment protocols of Covid-19, being a situation in continuous and rapid evolution. For this reason, I very much appreciated this manuscript in which the authors conduct a careful review of the current knowledge on the clinical use of a monoclonal antibody "Sarilumab" in moderate / severe stages of disease severity. I believe that the paper is well written and quite clear, with careful and detailed references to the molecular mechanisms of Interleukin 6 (IL-6) and its receptors, mainly represented by IL-6 R. We have also dealt with it thoroughly, with particular attention to the systemic and cutaneous manifestations of SARS-CoV-2, I would suggest the authors to read, study and quote the following papers, for a greater completeness and overview of the problem.

Cazzato G, Mazzia G, Cimmino A, Colagrande A, Sablone S, Lettini T, Rossi R, Santarella N, Elia R, Nacchiero E, Maruccia M, Marzullo A, Maiorano E, Giudice G, Ingravallo G, Resta L. SARS-CoV-2 and Skin: The Pathologist's Point of View. Biomolecules. 2021 Jun 4;11(6):838. doi: 10.3390/biom11060838. PMID: 34200112; PMCID: PMC8227624.
Sallenave JM, Guillot L. Innate Immune Signaling and Proteolytic Pathways in the Resolution or Exacerbation of SARS-CoV-2 in Covid-19: Key Therapeutic Targets? Front Immunol. 2020 May 28;11:1229. doi: 10.3389/fimmu.2020.01229. PMID: 32574272; PMCID: PMC7270404.
Kayesh MEH, Kohara M, Tsukiyama-Kohara K. An Overview of Recent Insights into the Response of TLR to SARS-CoV-2 Infection and the Potential of TLR Agonists as SARS-CoV-2 Vaccine Adjuvants. Viruses. 2021 Nov 18;13(11):2302. doi: 10.3390/v13112302. PMID: 34835108; PMCID: PMC8622245.
Cazzato G, Foti C, Colagrande A, Cimmino A, Scarcella S, Cicco G, Sablone S, Arezzo F, Romita P, Lettini T, Resta L, Ingravallo G. Skin Manifestation of SARS-CoV-2: The Italian Experience. J Clin Med. 2021 Apr 8;10(8):1566. doi: 10.3390/jcm10081566. PMID: 33917774; PMCID: PMC8068198.
Conti P, Ronconi G, Caraffa A, Gallenga CE, Ross R, Frydas I, Kritas SK. Induction of pro-inflammatory cytokines (IL-1 and IL-6) and lung inflammation by Coronavirus-19 (COVI-19 or SARS-CoV-2): anti-inflammatory strategies. J Biol Regul Homeost Agents. 2020 March-April,;34(2):327-331. doi: 10.23812/CONTI-E. PMID: 32171193.
Cazzato G, Colagrande A, Cimmino A, Cicco G, Scarcella VS, Tarantino P, Lospalluti L, Romita P, Foti C, Demarco A, Sablone S, Candance PMV, Cicco S, Lettini T, Ingravallo G, Resta L. HMGB1-TIM3-HO1: A New Pathway of Inflammation in Skin of SARS-CoV-2 Patients? A Retrospective Pilot Study. Biomolecules. 2021 Aug 16;11(8):1219. doi: 10.3390/biom11081219. PMID: 34439887; PMCID: PMC8392002.
Minor comments:

Line 19: please correct “SARS-CoV2” in “SARS-CoV-2”

Author Response

Dear Editor,

We appreciate you and the reviewers for giving us the opportunity to submit a revised draft of the manuscript and for your precious time in reviewing our paper.

We have carefully considered the comments and tried our best to address every one of them. We hope the manuscript after meticulous revisions meet your high standards. Additional constructive comments, if any, are welcomed. Below we provide the point-by-point responses.

All modifications in the manuscript have been highlighted with “Track changes” option on MS Word.

Reviewer 2

Dear Publisher and Authors,
I spent a lot of time reviewing this manuscript as, before proceeding with the check, I wanted to study and compare the most up-to-date treatment protocols of Covid-19, being a situation in continuous and rapid evolution. For this reason, I very much appreciated this manuscript in which the authors conduct a careful review of the current knowledge on the clinical use of a monoclonal antibody "Sarilumab" in moderate / severe stages of disease severity. I believe that the paper is well written and quite clear, with careful and detailed references to the molecular mechanisms of Interleukin 6 (IL-6) and its receptors, mainly represented by IL-6 R. We have also dealt with it thoroughly, with particular attention to the systemic and cutaneous manifestations of SARS-CoV-2, I would suggest the authors to read, study and quote the following papers, for a greater completeness and overview of the problem.

Cazzato G, Mazzia G, Cimmino A, Colagrande A, Sablone S, Lettini T, Rossi R, Santarella N, Elia R, Nacchiero E, Maruccia M, Marzullo A, Maiorano E, Giudice G, Ingravallo G, Resta L. SARS-CoV-2 and Skin: The Pathologist's Point of View. Biomolecules. 2021 Jun 4;11(6):838. doi: 10.3390/biom11060838. PMID: 34200112; PMCID: PMC8227624.
Sallenave JM, Guillot L. Innate Immune Signaling and Proteolytic Pathways in the Resolution or Exacerbation of SARS-CoV-2 in Covid-19: Key Therapeutic Targets? Front Immunol. 2020 May 28;11:1229. doi: 10.3389/fimmu.2020.01229. PMID: 32574272; PMCID: PMC7270404.
Kayesh MEH, Kohara M, Tsukiyama-Kohara K. An Overview of Recent Insights into the Response of TLR to SARS-CoV-2 Infection and the Potential of TLR Agonists as SARS-CoV-2 Vaccine Adjuvants. Viruses. 2021 Nov 18;13(11):2302. doi: 10.3390/v13112302. PMID: 34835108; PMCID: PMC8622245.
Cazzato G, Foti C, Colagrande A, Cimmino A, Scarcella S, Cicco G, Sablone S, Arezzo F, Romita P, Lettini T, Resta L, Ingravallo G. Skin Manifestation of SARS-CoV-2: The Italian Experience. J Clin Med. 2021 Apr 8;10(8):1566. doi: 10.3390/jcm10081566. PMID: 33917774; PMCID: PMC8068198.
Conti P, Ronconi G, Caraffa A, Gallenga CE, Ross R, Frydas I, Kritas SK. Induction of pro-inflammatory cytokines (IL-1 and IL-6) and lung inflammation by Coronavirus-19 (COVI-19 or SARS-CoV-2): anti-inflammatory strategies. J Biol Regul Homeost Agents. 2020 March-April,;34(2):327-331. doi: 10.23812/CONTI-E. PMID: 32171193.
Cazzato G, Colagrande A, Cimmino A, Cicco G, Scarcella VS, Tarantino P, Lospalluti L, Romita P, Foti C, Demarco A, Sablone S, Candance PMV, Cicco S, Lettini T, Ingravallo G, Resta L. HMGB1-TIM3-HO1: A New Pathway of Inflammation in Skin of SARS-CoV-2 Patients? A Retrospective Pilot Study. Biomolecules. 2021 Aug 16;11(8):1219. doi: 10.3390/biom11081219. PMID: 34439887; PMCID: PMC8392002.
Reply: Thank you for your precious suggestions. We found very interesting the articles you considered and we included them in the discussion section updating the references list.

Minor comments:

Line 19: please correct “SARS-CoV2” in “SARS-CoV-2”

Reply: We fixed the typo.

Reviewer 3 Report

In this review article, the Marino A. et al talked about the Sarilumab a drug which is already approved for rheumatoid arthritis.  They have mentioned that Sarilumab could be a better alternative treatment against COVID-19 because its capability to block IL-6R that would support to reduce excessive amount of cytokine release during COVID-19 infection.

The entire review article is interesting and well-written; However, I have some minor concerns that needs to be fulfilled.

  1. Author needs to mention either COVID-19 or COVID19 in entire review article. At some places it is written like COVID-19 and some places it is COVID19.
  2. Author should mention some negative feedback of Sarilumab after treatment in the patients from some other diseases or side effect also. Authors just concentrate only on the positive feedback direction.
  3. Authors needs to cite more references where they needed eg: Those have shown these above-mentioned points.
  4. This manuscript has several grammatical and typographical errors and needs thorough language editing.
  5. Figure 2, need to be correct with Figure 1 in figure legends.
  6. High DPI images need to be use in figure specifically.

Author Response

Dear Editor,

We appreciate you and the reviewers for giving us the opportunity to submit a revised draft of the manuscript and for your precious time in reviewing our paper.

We have carefully considered the comments and tried our best to address every one of them. We hope the manuscript after meticulous revisions meet your high standards. Additional constructive comments, if any, are welcomed. Below we provide the point-by-point responses.

All modifications in the manuscript have been highlighted with “Track changes” option on MS Word.

Reviewer 3

In this review article, the Marino A. et al talked about the Sarilumab a drug which is already approved for rheumatoid arthritis.  They have mentioned that Sarilumab could be a better alternative treatment against COVID-19 because its capability to block IL-6R that would support to reduce excessive amount of cytokine release during COVID-19 infection.

The entire review article is interesting and well-written; However, I have some minor concerns that needs to be fulfilled.

  1. Author needs to mention either COVID-19 or COVID19 in entire review article. At some places it is written like COVID-19 and some places it is COVID19.

Reply: We are sorry for the typos, we fixed them using the form “COVID-19”.

  1. Author should mention some negative feedback of Sarilumab after treatment in the patients from some other diseases or side effect also. Authors just concentrate only on the positive feedback direction.

Reply: Thank you for your suggestions. We added some extra lines about other adverse drug reactions and about DDI issue.

  1. Authors needs to cite more references where they needed eg: Those have shown these above-mentioned points.

Reply: As suggested we added other references within the text, as you pointed out. If you referred to specific lines, which we did not fixed, please let us know.

  1. This manuscript has several grammatical and typographical errors and needs thorough language editing.

Reply: Thank you for your valuable opinion. In order to correct the mistakes you found, the paper has been revised by a mother tongue colleague.

  1. Figure 2, need to be correct with Figure 1 in figure legends.

Reply: As kindly recommended we fixed the typo.

  1. High DPI images need to be use in figure specifically.

Reply: We use the formatting rules along with DPI standards indicated in the journal’s guidelines.